# Multisensory Stimuli, Restorative Effect, and Satisfaction of Visits to Forest Recreation Destinations: A Case Study of the Jhihben National Forest Recreation Area in Taiwan

**DOI:** 10.3390/ijerph20186768

**Published:** 2023-09-15

**Authors:** Yu-Jen Chiang

**Affiliations:** Department of Cultural Resources and Leisure Industries, The Teachers’ College, National Taitung University, Taitung 95092, Taiwan; kchiang@nttu.edu.tw

**Keywords:** multi-sensory experience, mental restoration, psychological benefits, forest service, green space, attention restoration theory

## Abstract

The perceived quality of the restorative environment of forest resources should be a key consideration for forest recreational areas in managing ecosystem services to provide health benefits to visitors. However, previous studies on utilizing forests as a resource for restorative services have rarely explored the simultaneous integration of on-site sensory experiences from a multisensory perspective or evaluated visitor satisfaction from a service-oriented standpoint. Therefore, this study aimed to understand the association among multisensory stimuli, perceived restorativeness, and satisfaction with visits to forest recreation areas and clarify the mediating role of perceived restorativeness in the relationship between multisensory stimuli and satisfaction. This study deployed a questionnaire and collected 458 valid responses from visitors to the Jhihben National Forest Recreation Area in Taiwan. Structural equation modeling was conducted to test the study hypotheses. The results indicated that visual sensations, auditory sensations, olfactory sensations, and tactile sensations had significant positive effects on perceived restorativeness and satisfaction and that perceived restorativeness also had a significant positive effect on satisfaction. Perceived restorativeness played a partial mediating role in this causal model. This study verified the psychological model of the relationships among a natural setting’s multisensory stimuli, perceived restorativeness, and satisfaction. In practice, the findings of this study offer guidance for forest recreation areas to develop strategies for ecological services, encompassing the establishment of restorative environmental management and visitor service management approaches.

## 1. Introduction

With the development of technology and urbanization, people’s lives are becoming increasingly isolated from nature. City living is associated with increased rates of stress, anxiety, and mental illness [1,2]. Improving mental health has become an important topic of discussion in the literature [3].

Attention restoration theory (ART), proposed by Kaplan and Kaplan in 1989 [1], is the basis of environmental psychology, which is often applied to explain the psychological benefits of environmental stimuli on people [4]. This theory emphasizes that in environments with restorative characteristics, attention recovery may lead to psychological benefits on people’s mental fatigue [1]. According to ART, a restorative environment that facilitates speedier recovery from attention fatigue has four environmental characteristics: being away (refers to stepping away from daily life and work; individuals can rejuvenate themselves through rest), fascination (related to the engaging qualities of the environment that hold a person’s interest and attention), compatibility (shows the degree to which an environment aligns with an individual’s personal inclinations and preferences), and extent (represents the environment allowing people to immerse and enjoy themselves in a continuous imagination and effortless exploration) [5,6]. Many previous studies have demonstrated that people in environments with restorative potential can recover from stress, increase their positive emotions, and reduce their negative emotions, which can promote happiness [1,4,7]. Compared with artificial environments, natural environments are believed to reduce negative emotions such as stress and anxiety, promote positive emotions, and even positively impact attention recovery and have other psychological benefits [8,9,10,11,12].

As an important component of natural environments, forests have a vital function in ecosystem services. Forests can absorb carbon dioxide, purify the air, mitigate climate change, protect natural environments [13,14], and produce negative ions, oxygen, and phytoncine. Therefore, an appropriate microclimate environment can effectively improve people’s health [10,15,16]. Lee et al. [17] surveyed visitors to Zhunan Protection Forest in Taiwan and found that visitors with a high perception of forest ecosystem services had a higher restorative state and more positive emotional value. As seen from the above research, the perceived quality of a restorative forest environment must be considered when providing ecosystem services. Due to the positive effects of forests on people’s health, Asian countries, such as Japan, the Republic of Korea, and Taiwan, have actively promoted forest therapy in recent years [18]. This shows that governments of various countries are attaching increasing importance to developing forests that can provide visitors with physical and mental health benefits.

People use multiple senses (sight, hearing, smell, taste, and touch) to experience their surroundings [19,20]; therefore, the environmental stimuli received by people’s senses create further psychological effects. Carlson [21] pointed out that natural environments are a suitable choice for multisensory appreciation in a sensory experience study on natural environments. Although sight is considered the most important sense for perceiving a natural environment, the effects of other senses, such as hearing and touch, on mental restoration in a natural environment should not be ignored [22,23]. Ulrich [24] pointed out that some sounds and smells in nature can affect people’s feelings and emotions. Research on the perception of restorative environments has increasingly focused on sound landscapes, smell landscapes, taste landscapes, and touch landscapes [25]. Song et al. [26] experimented with forest photos and forest sound recordings, and they found that single and combined visual or auditory stimuli could relax experimental subjects. Song et al. [27] conducted another experiment with forest photos and Hinoki cypress leaf oil, and they found that single or combined visual or olfactory stimuli could psychologically relax experimental subjects. Based on the above findings, people’s sensory stimuli in a natural environment have unique effects on their psychology, which is also associated with perceived restorativeness. Forest experiences mainly provide multisensory stimuli from a natural environment. Thus, it is important to study how to provide a restorative environment that people can feel in a forest.

Satisfaction has long been a major research theme in tourism [28]. According to the World Tourism Organization [29], consumer satisfaction is “a psychological concept that involves the feeling of well-being and pleasure that results from obtaining what one hopes or expects from an appealing product and/or service”. Tribe and Snaith [30] defined destination satisfaction as the gap between visitors’ evaluations of a destination’s attributes and their original expectations of these attributes. Bosque and Martín [31] believed that visitor satisfaction is a cognitive and emotional state experienced by individuals from their tourism experience. Thus, visitor satisfaction is a cognitive and emotional experience based on a visitor’s evaluation of a destination’s attributes.

Berto [32] pointed out that ART can explain people’s emotional responses to the restoration generated from their interaction with nature, such as their satisfaction and preference for landscapes. Therefore, satisfaction is based on whether the experience of environmental quality and environmental restoration of natural resources meets the needs of visitors and leads to their relaxation, positive emotions, and happiness [33]. Forest recreation and health promotion are important forest ecosystem services, and multisensory experiences are an important element in the planning, management, and marketing of perceived restorative environments [34]. Thus, understanding how visitors perceive the restorative quality of a forest environment through their senses and the resulting satisfaction is important in the study of psychological restoration.

Although grounded in the foundation of multisensory experiences and ART, understanding an environment that enables visitors to perceive a restorative quality through their senses and derive satisfaction holds importance in forest recreation marketing and environmental management. However, past studies on the association between sensorial dimensions and perceived restorativeness have conducted experiments and surveys using off-site simulations (images, recorded sounds, or other experimental settings) [35]. Few studies have investigated the association between perceived restorativeness and sight, hearing, smell, and touch on site; moreover, few studies have focused on the effect of perceived restorativeness on satisfaction. Therefore, the primary objective of this study is to survey the psychological model of multisensory stimuli, perceived restorativeness, and satisfaction. In addition, previous research has pointed out that perceived restorativeness plays a mediating role in the relationship between environmental experience and restorative outcomes [36,37]. Thus, this study further investigated the mediating role of perceived restorativeness in the relationship between the visual, auditory, olfactory, and tactile dimensions of satisfaction.

Academically, the results of this study could enrich and clarify research on the psychological model of the associations among multisensory stimuli, perceived restorativeness, and satisfaction. In addition, from a practical point of view, the results of this study could be used as a reference for forest resource management organizations to plan and manage recreation and psychological restoration services.

## 2. Theoretical Basis and Hypothesis Development

Grounded cognition theory emphasizes that individual sensory perceptions and cognition are not independent of each other, and the interaction between an individual and their external environment affects their cognition. Physical experiences affect people’s cognition. This is an inseparable part of the cognitive process. Therefore, people’s cognition is influenced by their interactions with external environments [38]. An individual’s cognitive responses can only be understood through their physical experiences. Traditionally, grounded cognition theory has been used to explain how multisensory stimuli play an important role in the generation of perceived restorativeness in natural environments [39,40]. Multisensory stimuli are the basis of the environmental restoration perceived by people [34], and the health benefits generated from exposure to nature are related to individuals’ sensory perceptions of their natural environments [41]. Multisensory stimuli are an important key to understanding perceived restorativeness and its health benefits. Therefore, this study explored the association between visitors’ multisensory stimuli and perceived restorativeness at the Jhihben National Forest Recreation Area using grounded cognition theory as the theoretical basis.

### 2.1. Multisensory Stimuli and Perceived Restorativeness

Environments with perceived restorativeness can help visitors reduce stress, evoke positive emotions, release negative emotions, and improve cognitive functions [42,43]. People’s perceived restorativeness is based on their sensory experiences [22]. Previous studies have pointed out that it is effective to use sensory information to trigger visitors’ experiences [44,45]. It is proposed that multisensory experiences are very important in the perceived restorativeness process [40,46].

The experience of a place originates from various sensory inputs in the destination environment, which trigger cognition, emotion, and place attachment [47]. Previous studies have pointed out that positive sensory experiences can bring positive emotions [48,49]. According to the findings of an empirical study on rural tourism in three villages in northern Portugal, visual, olfactory, and tactile experiences have significant effects on the generation of pleasant emotions [34], while vision, hearing, and smell have significant positive effects on relaxing people’s moods. Alvarsson, Wiens, and Nilsson [50] found that birdsong can enhance visitors’ pleasure while viewing images of tropical rainforests helps to reduce stress. Qiu et al. [51] pointed out that when the field features of sound and visual landscapes are combined, a natural environment has better restorative effects. Based on research results from two natural tourism sites in China and Australia, Qiu, Jin, and Scott [52] found that both sound and visual landscapes have positive effects on the perceived restorativeness of visitors. Henshaw [53] pointed out that positive smellscapes can bring restorative effects to cities, making inhabitants feel healthier and happier. Zhang et al. [54] surveyed the restorative effects of the multisensory perception of green spaces in Hangzhou Park, China, and found that vision and hearing have both direct and indirect relationships with mental restoration, while tactile sensations have an indirect relationship with mental restoration (via emotional mediation). Based on the abovementioned findings, this study proposed the following hypotheses:

**H1a.** *Visual sensations have significant effects on perceived restorativeness*.

**H1b.** *Auditory sensations have significant effects on perceived restorativeness*.

**H1c.** *Olfactory sensations have significant effects on perceived restorativeness*.

**H1d.** *Tactile sensations have significant effects on perceived restorativeness*.

### 2.2. Sensory Stimuli and Satisfaction

Satisfaction is an emotional response to an overall consumption experience [55]. It has been pointed out that perceived experience is the driver of visitor satisfaction regarding a destination [56,57]. A tourism experience that satisfies all five senses can help a destination establish a strong sensory relationship with visitors [58] and improve visitors’ satisfaction with their sensory experiences. Ai et al. [59] and Youssoufi et al. [60] pointed out that the visual and sound characteristics of a neighborhood environment can affect residents’ satisfaction with their place of residence. Gozalo [61] surveyed satisfaction with green spaces in Cáceres, a medium-sized city in the southwest of Spain, and the results showed that noise and air quality, cleanliness, esthetics, and protection are correlated with overall satisfaction with green spaces, and noise has the most significant effect on the overall satisfaction with green spaces. Ma et al. [62] conducted a survey in Beijing, China, and found that perceived air pollution and noise have significant direct effects on satisfaction. Some studies have pointed out that when gardening, touching the soil can bring emotional pleasure and satisfaction [63,64,65]. Based on the abovementioned research, this study proposed the following hypotheses:

**H2a.** 
*Visual sensations have significant effects on satisfaction.*


**H2b.** 
*Auditory sensations have significant effects on satisfaction.*


**H2c.** 
*Olfactory sensations have significant effects on satisfaction.*


**H2d.** *Tactile sensations have significant effects on satisfaction*.

### 2.3. Perceived Restorativeness and Satisfaction

Kim [66] surveyed visitors to a national park in the Republic of Korea and verified the relationships among perceived restorativeness, satisfaction, and loyalty. Using a survey of visitors to Chinese forest resorts, Chen et al. [67] examined the relevance of the perceived destination restorative qualities (PDRQs) scale in the context of Chinese culture. Fascination and compatibility can predict satisfaction and behavioral intention after visiting forest resorts. According to a survey of Chinese visitors visiting Thailand, Lu and Ampostira [68] found that mental getaways, compatibility, and fascination assessed based on the PDRQs scale have a significant effect on satisfaction. Based on the abovementioned research, this study proposed the following hypothesis:

**H3** . *Perceived restorativeness has significant effects on satisfaction*.

### 2.4. Mediating Role of Perceived Restorativeness

From the perspective of the five senses theory, the senses (sight, smell, taste, hearing, and touch) are the basis of people’s perceptual landscape, and these five senses are stimulated by different landscapes and form people’s experiences of their environment and contribute to further behavioral development [69]. Scopelliti et al. [70] pointed out that the perceived restorative quality of an environment plays a mediating role in the relationship between a natural environment and people’s psychological benefits. Based on the above literature, the sensory stimuli received from natural environments with restorative characteristics can generate restorative experiences and induce further healthy extensions of their experiential and cognitive benefits. Although there is no literature on the mediating role of perceived restorativeness in the relationship between visual, auditory, olfactory, and tactile sensations and satisfaction, it was previously pointed out that perceived restorativeness plays a mediating role in the relationship between green settings and quality of life [71], and the restorative effect mediates the relationship between soundscapes or visualscapes and quality of life [52]. Stedman [72] defined place satisfaction as a multidimensional summary judgment of the perceived quality of a setting, ranging from sociability to services and physical characteristics. Thus, perceived restorativeness may play a mediating role in the relationship between multisensory perception in a green setting and satisfaction. Furthermore, it has been shown from research on the direct relationships between variables that sight, hearing, smell, and touch are the predisposing factors in perceived restorativeness and that sight, hearing, smell, touch, and perceived restorativeness are also the predisposing factors in satisfaction. Based on the above-mentioned research, perceived restorativeness may play a mediating role in the relationship between sight, hearing, smell, touch, and visitor satisfaction.

Furthermore, based on the direct relationships among the variables mentioned above, it can be inferred that visual, auditory, olfactory, and tactile dimensions are antecedents to perceived restorativeness, and these dimensions, along with perceived restorativeness, are antecedents to satisfaction. In conclusion, drawing from the aforementioned descriptions, this study postulates that perceived restorativeness may serve as a mediating variable between visitor satisfaction and visual, auditory, olfactory, and tactile dimensions.

The conceptual framework of this study is shown in Figure 1.

## 3. Method

### 3.1. Study Site

The study site was the Jhihben National Forest Recreation Area, a famous national forest recreation area in eastern Taiwan, which is located in Beinan Township, Taitung County, Taiwan, adjacent to the renowned Zhiben Hot Springs area. It is 125 to 650 m above sea level, with an average annual temperature of 22 °C, and the total area is 110.8 hectares [73]. The ecological types and resources in this area are summarized based on Pan [74]. This area is characterized as a tropical monsoon forest. It features multilayered vertical forest communities and often showcases phenomena like epiphytic plants and buttress roots. Due to its rich plant diversity, this area also nurtures a wide range of animal resources, with birds being the most abundant, making it a famous birdwatching spot in Taitung. The tree species in forest recreation areas can be broadly categorized into two groups: native subtropical evergreen broadleaf trees, such as banyans, camphor trees, tung trees, and citrus trees like kumquats, and introduced tropical species like large-leaved mahogany, albizia, camphor, and pomelo.

According to the official website of the Jhihben National Forest Recreation Area [75], this area features four forest trails that allow visitors to enjoy forest bathing, admire the scenery, listen to the sounds of insects and birds, and breathe fresh air. Moreover, due to the diverse plant life in this forest recreation area, it boasts a rich array of animal resources, occasionally including unique Taiwanese species like Formosan macaques, Reeves’s muntjacs, and mikados. Additionally, this area offers various installations, including a “Monsoon through the Forest” zone, providing visitors with an auditory experience dominated by the wind, a water stream foot massage trail, and hot spring foot soaking pools, offering distinctive tactile experiences.

Overall, it is a place where visitors can engage with a natural environment that appeals to their senses of sight, sound, smell, and touch. This site also fits the criteria for achieving a state of restorativeness, as defined by Korpela and Hartig [76]. Human senses are stimulated by specific objects, such as the less complex natural landscape or the focal point among the landscape.

### 3.2. Research Instrument

The questionnaire used in this study focused on the three variables multisensory stimuli, perceived restorativeness, and satisfaction. The sensory landscape scale was mainly developed by referring to the studies by Zhang et al. [54] and Xu et al. [77]. The questionnaire contained nineteen questions, including six questions relating to visual sensations, five questions relating to auditory sensations, four questions relating to olfactory sensations, and four questions relating to tactile sensations.

To measure perceived restorativeness, this study primarily used a short version of the Perceived Restorativeness Scale (PRS) proposed by Berto [32]. Berto’s short version of the PRS is based on Korpela and Hartig’s (1996) PRS, with each dimension represented as a single question for a total of five questions. Additionally, inspiration was drawn from a study conducted by Tang et al. [78] in formulating these questions.

The satisfaction scale was modified by referring to a study by Taplin et al. [79] and included three questions. The primary reason for this choice is that Taplin et al.’s study focused on national park visitors who engage with natural environments, aligning closely with the nature of the participants in our study. These three items are also commonly utilized in overall satisfaction surveys for various location types, making it suitable to incorporate this three-item scale.

The respondents were asked to respond using a 7-point Likert scale, with answers ranging from 1 point (strongly disagree) to 7 points (strongly agree), representing the degree of agreement or disagreement.

The basic personal information of the respondents included gender, age, educational level, marital status, occupation, monthly income, residence, number of visits, and time spent in this area.

Before the formal draft and distribution of the questionnaire, an expert validity analysis was carried out. Two scholars with relevant backgrounds were invited to evaluate the sentences, the clarity of the question meanings, and the appropriateness of the wording in the questionnaire. The final questionnaire was revised based on the experts’ opinions (the questionnaire content is presented in Appendix A).

To ensure the reliability of the questionnaire, a pilot test was conducted before the formal test. The subjects of the pilot test were students who had been to the Jhihben National Forest Recreation Area in the past two months. For the pilot test, 100 questionnaires were distributed, and 91 valid questionnaires were obtained. The results showed that Cronbach’s α coefficients for each dimension were as follows: sight = 0.76; hearing = 0.86; touch = 0.88; perceived restorativeness = 0.79; and satisfaction = 0.9. Cronbach’s α coefficients were greater than 0.7 in all dimensions. According to Hair et al. [80], coefficients between 0.60 and 0.70 are acceptable, and values higher than 0.7 indicate high reliability. Therefore, all measurement items for each variable were retained without deletion in this study.

### 3.3. Sampling and Questionnaire Distribution

This study recruited visitors who were about to leave after completing activities in the Jhihben National Forest Recreation Area as research subjects. A convenience sampling method was used, and the questionnaires were distributed to visitors above the age of 20. To minimize potential sampling bias, the operational principles for the questionnaire survey in this study are outlined as follows: (1) The interviewers undergo training before conducting surveys. They are informed about the survey’s purpose and instructed to avoid personal subjective preferences when selecting interview subjects. (2) Regardless of the group size, only one voluntary participant is invited to respond. (3) Interviewers are stationed at both exits of the forest area to prevent sampling bias related to the survey location. (4) Each exit is staffed with two interviewers during each time period (morning and afternoon), maximizing the opportunity to engage with potential interviewees from every group. (5) The survey period used in this study takes into consideration both weekdays and weekends, with surveys conducted during both morning and afternoon time slots each week. (6) To enhance the willingness of visitors to fill out the questionnaire, interviewers explain the purpose of the survey to the participants. Additionally, interviewers offer small incentives to further boost the motivation for questionnaire completion.

The formal questionnaires were distributed at two exits of the Jhihben National Forest Recreation Area from 26 October 2022 to 30 November 2022 (8 a.m.–5 p.m.). During the survey period, the average temperature was 20 °C to 26 °C, which was more comfortable than that in July and August (summer) and January and February (winter). A total of 470 questionnaires were distributed, and 458 valid questionnaires were obtained.

### 3.4. Data Analysis

This study used SPSS 16.0 statistical software to conduct a descriptive analysis of the sample structure and reliability analysis. Linear structural equation modeling (SEM) and Amos 18 statistical software were used to perform the confirmatory factor analysis, calibrate the model parameters, verify the hypotheses, and verify the causal model and goodness of fit (GoF) of the overall model.

The mediation effect was analyzed using the bootstrap method, in which the direct and indirect effects of perceived restorativeness on the four sensory dimensions and satisfaction were analyzed. If the indirect effect was significant, it had a mediating effect; if the direct effect was significant, there was a partial mediating effect; and if the direct effect was not significant, perceived restorativeness had a full mediating effect [81,82].

## 4. Results

### 4.1. Profile of the Respondents

In terms of the respondents’ demographic characteristics (Table 1), 54.1% were female, and most respondents were married (56.6%). The respondents were mainly between the ages of 40 and 49 (26.6%), followed by the age group between 50 and 59 (23.1%). Most respondents had a college degree (64.0%), and many were service industry workers (23.1%). Most respondents earned between TWD 30,001 and 40,000 (22.5%) and TWD 40,001 and 50,000 (21.0%) per month. Most respondents lived in northern Taiwan (27.9%), followed by southern Taiwan (27.3%), and 65.1% of the respondents had visited the area for the first time. Most respondents spent one to two hours at the park (42.1%), followed by two to three hours (36.1%).

### 4.2. Assessment of the Measurement Model

Table 2 summarizes the confirmatory factor analysis of the measurement model. The results of the skewness and kurtosis analysis showed that skewness ranged between −1.156 and −0.252, while kurtosis ranged between −0.622 and 1.670, meeting the assumptions that the skewness coefficient should be less than 3 and the kurtosis coefficient should be less than 10 [83]. The standardized loads of all dimension items ranged from 0.616 to 0.866 and composite reliability (CR) ranged from 0.802 to 0.896, meeting the standard of a standardized load greater than 0.5 and a CR greater than 0.7, as suggested by Hair et al. [84]. The average variance extracted (AVE) for all constructs ranged from 0.431 to 0.742. Previous research has suggested that the AVE value should be higher than 0.5; however, an AVE value above 0.36 is still an acceptable standard [85].

### 4.3. Results of the Discriminant Validity Test

Hair et al. [86] pointed out that the square root of the AVE for each dimension should be greater than the correlation coefficient of each dimension. As shown in Table 3, the square root of the AVE for each dimension is greater than the correlation coefficient of each dimension, proving that the measurement model in this study has discriminative validity. In addition, the results of this study are in line with those of Gaski and Nevin [87], who suggested that the correlation coefficient between any dimensions should be lower than one and that the correlation coefficient between any two dimensions should be lower than the two test criteria for Cronbach’s α reliability for these dimensions.

### 4.4. Analysis of the Overall Model’s Goodness of Fit

In this study, the overall structural model fit was as follows: χ^2^ = 640.197; df = 309; χ^2^/df = 2.072; GFI = 0.905; RMSEA = 0.048; CFI = 0.944; SRMR = 0.042; TLI = 0.936; PNFI = 0.790; and PCFI = 0.831. All these values were in line with the proposed values of Hair et al. [88]. The AGFI value (0.883) did not meet the standard of 0.90; however, Etezadi-Amoli and Farhoomandy [89] suggested that the AGFI value should be relaxed and that any result above 0.8 can be acceptable.

### 4.5. Tests of the Hypotheses

As shown in Figure 2, all nine hypotheses are supported, including H1a: visual sensations → perceived restorativeness; H1b: auditory sensations → perceived restorativeness; H1c: olfactory sensations → perceived restorativeness; H1d: tactile sensations → perceived restorativeness; H2a: visual sensations → satisfaction; H2b: auditory sensations → satisfaction; H2c: olfactory sensations → satisfaction; H2d: tactile sensations→ satisfaction; and H3: perceived restorativeness → satisfaction. In terms of the path coefficients, auditory sensations have higher impacts on perceived restorativeness, followed by olfactory sensations. Auditory sensations have higher impacts on satisfaction, followed by tactile sensations.

### 4.6. Mediation Analysis

The bootstrapping method proposed by Preacher and Hayes [81] is widely considered suitable for testing the effects of mediation. Therefore, the bootstrapping function in AMOS20 was utilized in this study, and the overall effect was examined. The direct effect and indirect effect were tested (the sample number was set as 2000, and the bias-corrected bootstrap method was used for obtaining the 95% confidence interval) to understand the mediating effect of perceived restorativeness on the relationship between visual sensations, auditory sensations, olfactory sensations, and tactile sensations and satisfaction.

As shown in Table 4, the results of the mediation analysis with visual, auditory, olfactory, and tactile sensations as the independent variables, perceived restorativeness as the mediating factor, and satisfaction as the dependent variable show that the confidence intervals for the indirect effects of visual, auditory, olfactory, and tactile sensations do not include zero and that their direct and total effects also do not include zero, indicating partial mediation in this model. Therefore, perceived restorativeness plays a partial mediating role in the relationship between visual, auditory, olfactory, and tactile sensations and satisfaction.

## 5. Discussion

This study aimed to understand the relationships between multisensory stimuli, perceived restorativeness, and satisfaction with visits to forest recreation areas. The results showed that sight, hearing, smell, and touch had significant positive effects on perceived restorativeness and satisfaction and that perceived restorativeness had significant positive effects on satisfaction. The results of the mediation analysis showed that perceived restorativeness played a partial mediating role in the relationship between multisensory stimuli and satisfaction. The findings of this study are consistent with those of Grahn and Stigsdotter [22], who indicated that the perceived surrounding environment connects people’s senses and health. These results also agree with those reported by Agapito et al. [34], who stated that physical sensations generated by sensory experiences determine visitors’ evaluations of their travel experiences.

Sight, hearing, smell, and touch all had significant positive effects on perceived restorativeness. These results suggest the importance of providing multisensory stimuli as part of the perceived restorativeness process in natural environments [39,40,41]. These findings are consistent with the views of previous studies that sensory experiences have a direct effect on mental restoration, including experiences such as viewing natural environments [23], listening to the sounds of nature [22], receiving forest-related visual and olfactory stimuli [27], tactile experiences in water [90], and walking barefoot [91]. These results are also similar to previous findings showing that the indirect simulation of forest experiences based on sight, hearing, smell, and touch have significant benefits in terms of improving people’s positive emotions [92] and that the visual, auditory, olfactory, and tactile experiences of visitors have significant effects on positive emotions [34].

The results of this study suggested that multisensory stimuli have a significant positive effect on satisfaction, indicating that the multisensory experiences of visitors help promote a positive experience. These results are similar to the findings of Kankhuni and Ngwira [93], who showed that visitors’ natural soundscape perceptions significantly affect satisfaction. The results are also consistent with those of Shao and Lin [94], who stated that the visual, auditory, and tactile sensations of visitors visiting towns with forest characteristics significantly affect the quality of their experiences. Moreover, the results echo the findings of Gozalo et al. [61], who showed that noise and air quality are significantly correlated with the overall satisfaction of users of urban green spaces.

Compared with other sensory stimuli, visual sensations usually play a dominant role [95]. The findings of this study are different from those of previous studies that emphasize the relationship between vision and attention restoration [96]. According to the path coefficients, auditory sensations have the highest effects on perceived restorativeness and satisfaction, while visual sensations have the lowest effects on perceived restorativeness and satisfaction. These results are similar to those of Zhang et al. [54], who stated that auditory sensations have the greatest effects (followed by visual and tactile sensations) on the mental restoration of visitors visiting Guangzhou City Park in China. The results of this study are also consistent with those of Hedblom et al. [97], who stated that smells may be better at reducing stress than visual stimuli. This study posits that in contrast with the environment of Taitung, there is a considerably higher level of noise and air pollution due to the predominant origin of visitors to the Jhihben National Forest Recreation Area from more urbanized regions in western Taiwan. Consequently, compared with the visually appealing green spaces, the sound of insects and birds in nature, good air quality, pleasant climate, and proximity to water in Taitung may serve as more prominent sources of psychological benefits for visitors. These sensory attributes likely play a significant role in generating visitors’ psychological benefits and contributing to their satisfaction with the environment.

## 6. Practical Implications

It is suggested that the Jhihben National Forest Recreation Area first establish a sensory resource database that can be used as the basis for management planning to promote visitors’ stress recovery and satisfaction in the future. In this study, the effects of auditory sensations on perceived restorativeness and satisfaction were higher than that of the other three sensations. Thus, it is suggested that the administration of recreation areas draw maps of the auditory landscape for all four seasons so that visitors can feel the changes in different seasons (e.g., different tree species, different terrains, or different places where birds or insects appear). “Nature Quiet” is a global trend aiming to provide forest therapy. It is suggested that quiet areas and trails be planned for visitors to feel nature in the quiet.

Regarding the dimension of olfactory sensations, dense forests in an area can provide visitors with fresh air vastly different from air in the city. It is suggested that visitors be encouraged to perform deep-breathing exercises and use the smell landscape map of the Jhihben National Forest Recreation Area so that they can experience the fragrance of different natural environments. Even small areas with local fragrant plants can be provided so visitors can have unforgettable olfactory memories and sensory experiences.

Regarding the dimension of tactile sensations, it is suggested that trail maintenance focus on variability in natural materials (e.g., seasonal trails containing fallen leaves) and strengthening the function of areas near waterbodies to increase the opportunity for tactile experiences. Previous studies have found that walking barefoot can contribute to psychological restoration. Therefore, it is suggested that barefoot trails be set up in the Jhihben National Forest Recreation Area using suitable materials.

Regarding the dimension of visual landscapes, although there are several high points in the Jhihben National Forest Recreation Area where visitors can enjoy a panoramic view, the visual perspective is not ideal because there are many trees of different heights along the walkways. It is suggested that branches be trimmed or tidied up properly without affecting the ecology so visitors can admire the green landscape along the way and improve their visual experience. Related facilities, such as the visitor center, should be strengthened with the greening of building facades, and natural materials should be used for related facilities along the walkways (such as seats and gazebos) to harmonize human-made facilities with the natural environment.

## 7. Conclusions

This study verified the psychological model of the relationships among multisensory stimuli, perceived restorativeness, and satisfaction. The four sensory stimuli (sight, hearing, smell, and touch sensations) considered in this study all affect perceived restorativeness and satisfaction, and perceived restorativeness had significant positive effects on satisfaction. In comparison, it was found that the influence of auditory sensations on both perceived restorativeness and satisfaction was the highest, while the influence of visual sensations was the lowest. As evident from the above, visitors’ sensory stimuli impact their mental recovery in a forest environment. In addition to the visual aspect, the roles of auditory, olfactory, and tactile sensations should also be valued. Perceived restorativeness plays a partial mediating role in the relationship between sensory stimuli and satisfaction. Therefore, promoting the multisensory aspect of nature experiences is crucial for improving visitors’ psychological restorations and their satisfaction with the service quality.

The results of this study demonstrate the relationships among visitors’ multisensory stimuli, perceived restorativeness, and satisfaction when visiting forest recreation areas. However, this study had certain research limitations. First, seasons cause people to feel different sensory sensations. However, as the time of the questionnaire distribution in this study was limited to one month of data collection in autumn, the status of other seasons could not be known. Second, this study was conducted for only one month, thus limiting the understanding of visitors’ experiences in other months. Additionally, the local daily environmental conditions, such as light, humidity, temperature, and even the number of people encountered, varied to some extent. These differences might have affected respondents’ answers.

Moreover, although the results of this study demonstrate the effects of visual, auditory, olfactory, and tactile sensations, as well as perceived restorativeness, on overall visitor experience, the results also show that auditory sensations have a relatively greater contribution to mental restoration and satisfaction than the other three sensations. In future research on the psychological benefits of forest recreation areas, surveys based on additional multisensory perspectives will be needed to understand the effects of individual senses and empathy.

## Figures and Tables

**Figure 1 ijerph-20-06768-f001:**
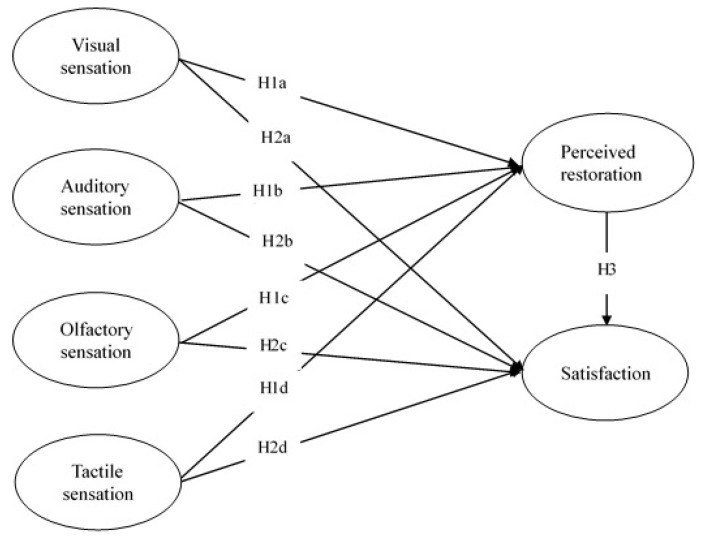
Proposed research model.

**Figure 2 ijerph-20-06768-f002:**
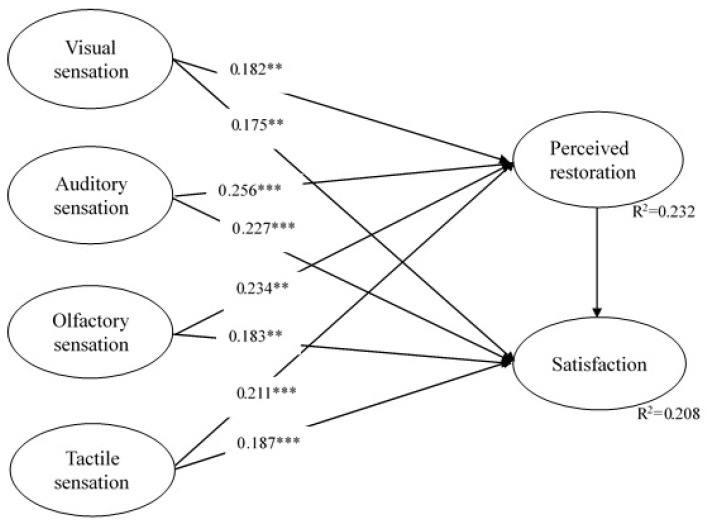
The estimated structural model. Note: ** *p*-value < 0.01, and *** *p*-value < 0.001.

**Table 1 ijerph-20-06768-t001:** Profiles of the respondents.

Characteristics		Frequency	Percentage
Gender	Male	210	45.9
	Female	248	54.1
Age (years)	20–29	87	19.0
	30–39	89	19.4
	40–49	122	26.6
	50–59	106	23.1
	60 (and above)	54	11.8
Educational level	Senior high school (and below)	108	23.6
	College degree	293	64.0
	Master’s degree (and above)	57	12.4
Marital status	Married	259	56.6
	Single	199	43.4
Occupation	Military personnel, civil servants, and teachers	65	14.19
	Service industry	106	23.14
	Industrial	67	14.63
	Commerce	71	15.50
	Freelance	41	8.95
	Homemakers	30	6.55
	Students	46	10.04
	Others (retired, unemployed, etc.)	32	6.99
Monthly income (TWD)	20,000 (and below) (USD 654 and below)	59	12.9
	20,001–30,000 (USD 654–981)	64	14.0
	30,001–40,000 (USD 981–1309)	103	22.5
	40,001–50,000 (USD 1309–1636)	96	21.0
	50,001–60,000 (USD 1636–1963)	76	16.6
	60,001 (and above) (USD 1936 and above)	60	13.1
Residence	Northern Taiwan	128	27.9
	Central Taiwan	109	23.8
	Southern Taiwan	125	27.3
	Eastern Taiwan and others	96	21.0
Number of visits	First time	298	65.1
	Second time	76	16.6
	Third time	30	6.6
	Fourth time	28	6.1
	Fifth time (and above)	26	5.7
Time spent in the area	Less than 1 h	25	5.5
	1–2 h	193	42.1
	2–3 h	165	36.0
	Over 3 h	75	16.4

**Table 2 ijerph-20-06768-t002:** Confirmatory factor analysis of the measurement model.

Factors	Average	Skewness	Kurtosis	SFL (λ)	C.R	AVE
Visual sensations					0.818	0.431
1. Lush green vegetation	6.19	−0.617	0.080	0.617 ***		
2. Diversified botanical landscapes	6.07	−0.556	−0.121	0.616 ***		
3. Not crowded	6.18	−0.740	−0.179	0.625 ***		
4. Diverse animal landscapes (such as birds, butterflies, and monkeys)	6.16	−0.703	0.078	0.815 ***		
5. Coordinated human-made facilities and natural landscapes (roads, pavilions, seats, etc.)	5.98	−0.539	−0.081	0.621 ***		
6. Places with wide views	5.88	−0.349	−0.452	0.619 ***		
Auditory sensations					0.837	0.511
1. Pleasant natural sounds	6.38	−0.966	0.588	0.847 ***		
2. Harmonious natural sounds	6.17	−0.586	−0.262	0.645 ***		
3. No traffic noise	6.38	−1.019	0.298	0.688 ***		
4. Less human activity noise	6.30	−1.055	0.640	0.616 ***		
5. Quiet spaces	6.26	−0.963	0.214	0.753 ***		
Olfactory sensations					0.802	0.506
1. Fresh air	6.44	−1.076	0.665	0.717 ***		
2. Fragrant vegetation (trees, flowers, soil, etc.)	6.27	−1.156	0.786	0.674 ***		
3. No artificial odors	6.32	−1.075	0.664	0.828 ***		
4. No irritating or pungent smells	6.25	−0.750	−0.283	0.609 ***		
Tactile sensations					0.844	0.577
1. Comfortable temperature	5.91	−0.653	0.061	0.835 ***		
2. Gentle wind	5.66	−0.252	−0.622	0.804 ***		
3. Suitable road pavement	5.79	−0.441	−0.288	0.685 ***		
4. Enjoyment from touching the water	5.91	−0.605	−0.318	0.705 ***		
Perceived restorativeness					0.830	0.496
1. This area is away from everyday demands and a place where I can relax and think about things that interest me (being away).	6.21	−1.129	1.670	0.845 ***		
2. This area is fascinating. It is large enough for me to discover and be curious about things (fascination).	6.00	−0.872	0.649	0.712 ***		
3. It is easy to orient, move around, and do what I like in this area (compatibility).	6.10	−0.795	0.917	0.636 ***		
4. This area feels like a world of its own, and it is easy for me to move around it (scope).	5.76	−0.520	0.075	0.617 ***		
5. This area has activities, services, and attributes that are well-ordered and organized (coherence).	5.80	−0.770	1.219	0.690 ***		
Satisfaction					0.896	0.742
1. I am satisfied with my decision to visit this area.	6.01	−0.670	−0.124	0.866 ***		
2. A visit to this area gives me exactly what I need.	6.10	−1.028	1.362	0.858 ***		
3. Overall, I am satisfied with my visit to this area.	6.17	−0.812	1.100	0.861 ***		

Note: SFL = standardized factor loading; AVE = average variance extracted; and CR = composite reliability; *** = *p* < 0.001.

**Table 3 ijerph-20-06768-t003:** Correlation coefficients of the dimensions.

Construct (Item Number)	V	A	O	T	PR	SAT
**V (6)**	**0.656**					
**A (5)**	0.537 **	**0.715**				
**O (4)**	0.468 **	0.545 **	**0.711**			
**T (4)**	0.426 **	0.346 **	0.489 **	**0.760**		
**PR (5)**	0.475 **	0.506 **	0.498 **	0.466 **	**0.711**	
**SAT (3)**	0.555 **	0.562 **	0.559 **	0.514 **	0.572 **	**0.862**
**Cronbach’s α**	0.814	0.829	0.786	0.843	0.828	0.893

Note 1: V = visual; A = auditory; O = olfactory; T = tactile; PR = perceived restorativeness; and SAT = satisfaction. Note 2: The diagonal values (in bold) represent the square root of the AVE. The value below the diagonal is the normalized (Pearson) correlation coefficient. Note 3: ** *p* < 0.01.

**Table 4 ijerph-20-06768-t004:** Indirect, direct, and total effects.

Effect	Relationship Path	β	95% CI (Low, High)
Indirect effect	visual sensations → perceived restorativeness → satisfaction	0.038 **	(0.005, 0.095)
Direct effect	visual sensations → satisfaction	0.175 **	(0.054, 0.311)
Total effect	visual sensations → satisfaction	0.213 **	(0.095, 0.341)
Indirect effect	auditory sensations → perceived restorativeness → satisfaction	0.053 **	(0.009, 0.133)
Direct effect	auditory sensations → satisfaction	0.227 **	(0.079, 0.377)
Total effect	auditory sensations → satisfaction	0.280 **	(0.147, 0.413)
Indirect effect	olfactory sensations → perceived restorativeness → satisfaction	0.048 *	(0.009, 0.119)
Direct effect	smelling → satisfaction	0.183 *	(0.002, 0.352)
Total effect	smelling → satisfaction	0.232 *	(0.056, 0.395)
Indirect effect	tactile sensations → perceived restorativeness → satisfaction	0.044 **	(0.008, 0.098)
Direct effect	touch → satisfaction	0.187 **	(0.077, 0.295)
Total effect	touch → satisfaction	0.230 **	(0.117, 0.340)

Note 1: bootstrapped N = 2000; *p* = Bollen–Stine bootstrapped *p*. Note 2: * *p* < 0.05; ** *p* < 0.01. Note 3: β = coefficient; 95% CI = bias-corrected 95% confidence interval.

## Data Availability

The data that support the findings of this research are available from the author, Y.-J.C., upon reasonable request.

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
