# Peer review of "Multisensory Stimuli, Restorative Effect, and Satisfaction of Visits to Forest Recreation Destinations: A Case Study of the Jhihben National Forest Recreation Area in Taiwan"

_ijerph, 2023, doi:10.3390/ijerph20186768_

Round 1
Reviewer 1 Report
The research topic appears to be of interest and would be suitable for publication in IJERPH (ISSN 1660-4601). This manuscript examines the content of "Multisensory Stimuli, Restorative Effect, and Satisfaction of Visit to Forest Recreation Destinations: A Case Study of the Jhihben National Forest Recreation Area in Taiwan" (ijerph-2526224), which aimed to understand the association among multisensory stimuli, perceived restorativeness, and satisfaction of visits to forest recreation destinations as well as to clarify the mediating role of perceived restorativeness in the relationship between multisensory stimuli and satisfaction.
This study has some technical and structural issues that should be addressed during the revision process.
Following are the details of the comments.
1. As this study is based on a questionnaire, this questionnaire is not included in the study, which makes it not scientifically valid. This study would lose all its impact and significance if the entire questionnaire was not provided in English as a supplementary file.
2. The manuscript should be extensively revised in terms of language and grammar.
Title
3. Visit or visits? “Multisensory Stimuli, Restorative Effect, and Satisfaction of Visits to Forest Recreation Destinations: A Case Study of the Jhihben National Forest Recreation Area in Taiwan”
Abstract
4. The background opening sentence is missing.
5. The research gaps statement, which is expected in the second sentence, is missing.
6. The policy recommendation, which is expected as the last sentence, is missing.
Keywords
7. The keywords matching the title should be replaced. Currently, the keywords used in this study are not sufficient to allow other readers to search for the study.
Introduction
8. Despite the fact that the author clearly outlined the research hypothesis, the study aims should also be stated.
Materials and Methods
9. The facts provided on the study site should be accompanied by supporting literature.
Results
10. Results are described in detail in the result section.
Discussion
11. This study lacks information regarding its limitations.
Conclusion
12. It is necessary to revise and clarify the conclusion section and remove the reference.
13. The manuscript contains some information that requires supporting literature. A review of the suggested or other literature supporting the content is recommended.
Extensive editing of English language required.
Reviewer 2 Report
The manusript has an interesting theme on the relationship among sensory stimuli, restoration and satisfaction.
1. This study based on the data from the survey of receationists in Jhihben National Forest Recreation Area. Therefore, sampling method is a very important tool for the study. The manuscript needs to specify how the questionnaire were adminstered. What is a convenience sampling method? Can this method secure representability of visitors?
2. In this study you employed PRS developed by Berto and Tang, et. al. You mentioned five questions were included. Did you used whole five questions from the original scale? If you delected some of them, why?
3. Same to the satisfaction scale
4. The sensory stimuli are influned by characteristics of forests, such as species, type of forests, etc. In the study area, characteristics of forest should be explained.
Round 2
Reviewer 1 Report
The revised manuscript has been significantly improved.